# Characterization, Conservation, and Breeding of Winter Squash (*Cucurbita moschata* Duchesne): Case Study of the Collection Maintained at the Federal University of Viçosa Vegetable Germplasm Bank

**DOI:** 10.3390/plants14152317

**Published:** 2025-07-27

**Authors:** Derly José Henriques da Silva, Ronaldo Silva Gomes, Ronaldo Machado Júnior, Cleverson Freitas de Almeida, Rebeca Lourenço de Oliveira, Dalcirlei Pinheiro Albuquerque, Santina Rodrigues Santana

**Affiliations:** 1Agronomy Department, Universidade Federal de Viçosa, Viçosa 36570-900, MG, Brazil; derly@ufv.br (D.J.H.d.S.); cleverson.almeida@ufv.br (C.F.d.A.); dalcirlei.albuquerque@ufv.br (D.P.A.); 2Department of Agricultural Sciences, Instituto Federal do Ceará, Crateús 63708-260, CE, Brazil; 3Agronomy Department, Universidade Federal de Uberlândia, Monte Carmelo 38500-000, MG, Brazil; ronaldo.juniior@ufv.br; 4Agronomy Department, Universidade Federal Rural do Rio de Janeiro, Campus dos Goytacazes 28020-560, RJ, Brazil; rebeca.oliveira@ufv.br; 5Department of Fisheries and Aquaculture Engineering, Universidade Federal de Rondônia, Presidente Médice 76916-000, RO, Brazil; rsant.1@hotmail.com

**Keywords:** carotenoid, cucurbit, fatty acid profile, germplasm collection, oleic acid, plant size, pumpkin

## Abstract

Winter squash (*Cucurbita moschata* Duchesne.) is a vegetable of high socioeconomic importance owing to the nutritional quality of its fruits, seeds, and seed oil. This study aims to review the main aspects related to the characterization, conservation, and breeding of *C. moschata*, emphasizing the studies with *C. moschata* accessions maintained by the Vegetable Germplasm Bank of the Federal University of Viçosa (BGH-UFV). Studies on *C. moschata* germplasm have reported high variability, particularly in Brazil. Currently, Brazil maintains six *Cucurbita* germplasm collections, kept in research and teaching institutions. The BGH-UFV collection, one of the largest in the country, contains approximately 350 accessions of *C. moschata*, mostly landraces collected from all over Brazil. Studies characterizing this germplasm have identified promising genotypes as sources of alleles for increasing the carotenoid content in the fruit pulp and oleic acid content in the seed oil. As part of a breeding program to increase seed oil productivity and improve the oil profile, studies with the BGH-UFV germplasm have identified *C. moschata* genotypes with seed oil productivity of up to 0.27 t ha^−1^ and accessions producing oil with high oleic acid content (21 to 28%). The genetic breeding program of *C. moschata* conducted at the UFV has prioritized the development of compact growth habit genotypes to reduce plant spacing and increase seed and oil productivity. The works involving the collection of *C. moschata* maintained by the BGH-UFV corroborates the importance of this germplasm as a source of alleles for improving seed oil productivity and the oil profile.

## 1. Introduction

Winter squash (*Cucurbita moschata* Duchesne) is cultivated worldwide and holds significant socioeconomic importance owing to the nutritional quality of its fruits, seeds, and seed oil. Initially, its cultivation was mainly intended for the production of fruits, which are sources of a series of components essential for human nutrition and health. The fruit pulp is rich in carotenoids, such as *α* and *β*-carotene [1,2,3], which are the main precursors of vitamin A and have pronounced antioxidant activity. Additionally, it is an excellent source of minerals, such as K, Ca, P, Mg, and Cu [3,4].

*C. moschata* is a vegetable belonging to the Cucurbitaceae family and the *Cucurbita* genus. It is a diploid species with 40 chromosomes organized in pairs (2n = 2x = 4O). Characteristically, plants of the *Cucurbita* genus have an annual cycle and indeterminate growth habit. They have herbaceous-prostrate stem, dark green, with tendrils and adventitious roots that help them anchor. Plants of this genus are monoic, with a predominance of male flowers.

The production of *C. moschata* seed oil for human consumption represents an excellent opportunity in the production of this vegetable. The oil from its seeds contains approximately 75% unsaturated fatty acids with a high content of oleic acid, a monounsaturated fatty acid [5,6,7]. Therefore, the seed oil of *C. moschata* constitutes an excellent substitute for lipid sources that are harmful to human health, such as those with high levels of saturated fatty acids.

Studies have highlighted that the seeds and seed oil of *C. moschata* contain high levels of antioxidant components, such as vitamin E and carotenoids [8,9], which, in addition to being beneficial to human health, protect the oil against oxidative processes. In addition to the nutritional aspects of its seed oil, *C. moschata* has a seed productivity of up to 0.58 t. ha^−1^ and a seed oil content of up to 49% [5,10,11], corroborating the high potential of this vegetable for oil production. In addition, *C. moschata* is cultivated worldwide, together with other *Cucurbita* species, such as *C. pepo* and *C. maxima*, with a cultivated area and production close to 1.5 million hectares and 23.6 million tons, respectively [12]. In Brazil, the combined cultivated areas and production of *C. moschata*, *C. maxima*, and *C. pepo* are approximately 78.67 thousand hectares and 417.83 tons, respectively, confirming their socioeconomic importance.

*Cucurbita moschata* is believed to have been first domesticated in Latin America, with Colombia corresponding to the primary center of diversity of the species. In Brazil, the germplasm of *C. moschata* shows remarkable variability, reflecting the adaptation of this species to different soil and climatic conditions found in the country. To conserve and uti-lize the genetic diversity of vegetables such as *C. moschata*, the Vegetable Germplasm Bank of the Federal University of Viçosa (BGH-UFV) maintains a collection of approximately 350 accessions of *C. moschata*, one of the largest collections of this species in Brazil [13].

This study aims to review the main aspects related to the characterization, conservation, and breeding of *C. moschata*, emphasizing the studies with *C. moschata* accessions maintained by the BGH-UFV. The results reveal that the assessment of this germplasm has led to the identification and use of accessions with resistance to important phytopathogens of the crop and accessions with high potential for use in improving the nutritional quality of fruit pulp and seed oil. The findings corroborate the importance of continuing studies involving the evaluation and use of the *C. moschata* germplasm from BGH-UFV.

## 2. Domestication and Genetic Diversity

Archaeological evidence suggests that the domestication of *C. moschata* occurred in Latin America. This evidence demonstrates that this species was present in Latin America before its colonization and appears to have been essential to the diet of the native people of this continent, dating back to the regions of Colombia and Ecuador [14,15,16]. In line with this, [17] highlighted that there is a consensus that Colombia is the primary center of diversity of *C. moschata* and that the cultivation of this crop in parts of the American continent precedes the pre-Columbian era. With the process of dispersal, the cultivation of *C. moschata* spread to different regions of Latin America and was introduced to North America and Asia, with its cultivation now carried out worldwide.

Studies on the *C. moschata* germplasm commonly report high variability in agromor-phological and molecular traits [11,18,19], as demonstrated by the morphological variation among fruits (Figure 1). In addition to being allogamous, which favors the occurrence of natural hybridization in *C. moschata* and increases its variability; anthropogenic action appears to have significantly contributed to the variability of this species. According to [11], the selection process practiced over time by populations involved in the cultivation of this vegetable, which is associated with the frequent exchange of seeds between populations, increased its variability.

All accessions shown in Figure 1 belong to the group of dry pumpkin cultivars; a group known in the Brazilian market for its large fruits, weighing up to 15 kg, with thick and hard skin and firm pulp.

The agromorphological variability of *C. moschata* is also reflected in its agronomic characteristics, such as cycle duration, fruit shape and mass, fruit productivity, carotenoid content, and seed mass per fruit [11,20].

## 3. Brazilian Germplasm of *C. moschata*

In Brazil, *C. moschata* is cultivated across almost the entire territory and has adapted to a wide ecological range, including different soil and climatic conditions. As a result, the germplasm of this species in the country shows high variability, thus constituting an important source for the genetic improvement of this vegetable and other *Cucurbits*.

Brazil currently maintains six collections of *Cucurbits* germplasm, resulting from a long collection period and numerous expeditions. Most of these collections are maintained by research units of the Brazilian Agricultural Research Corporation (EMBRAPA), such as EMBRAPA Climate Temperado, EMBRAPA Semiárido, EMBRAPA Hortaliças, and EMBRAPA Recursos Genéticos e Biotecnologia. *Cucurbits* collections are also maintained by educational institutions, including the Federal University of Viçosa (UFV) and the Agronomic Institute of Campinas (IAC). Except for the long-term preserved collection maintained by EMBRAPA Recursos Genéticos e Biotecnologia, the other *Cucurbits* collections are medium-term and are maintained in active germplasm banks. The accessions of *C. moschata* maintained in these collections total approximately 3600 [21].

Founded in 1966, the BGH-UFV maintains approximately 350 accessions of *C. moschata*, mostly landraces, from a collection period spanning more than five decades and covering all Brazilian territory [22]. These authors reported that most of this germplasm originated from intermittent collections carried out between 1960 and 1990. These authors reported that BGH-UFV maintained approximately 6500 accessions, comprising families such as Solanaceae (44.21%), Leguminosae (16.83%), Cucurbitaceae (15.70%), and others (23.26%). It is assumed that the collection of *C. moschata* maintained at BGH-UFV constitutes a substantial sample of the Brazilian germplasm, corresponding to one of the largest collections of this species in the country [13].

Most of the *C. moschata* collection maintained at BGH-UFV have already been characterized in terms of the agromorphological aspects and chemical-nutritional characteristics of the fruits, seeds, and seed oil. Studies involving this characterization have allowed the identification of accessions with resistance to an important phytopathogen of the crop, such as the *Zucchini yellow mosaic virus* (ZYMV) [23]; and accessions for use as parents to improve chemical-nutritional aspects, such as increasing the carotenoid content in the fruit pulp and the oleic acid content in the seed oil, aiming at the production of fruits and oil with higher chemical-nutritional quality [6,11,20,24]. To date, studies with *C. moschata* germplasm of BGH-UFV have prioritized the agromorphological evaluation of accessions, a crucial assessment in the production of this vegetable. Studies such as that by Sobreira [6] and Gomes et al. [11] provide detailed assessment of variability of the collection based on agromorphological traits. After the agromorphological evaluation of this collection is completed, the aim is also to perform its assessment at the DNA level.

Corroborating the importance of the BGH-UFV *C. moschata* collection, [11] highlighted that the use of this germplasm as a source of genes for the improvement of this species, together with the possibility of elucidating the genetic mechanisms of important production parameters, thus emphasizing the importance of continuing studies with this germplasm.

## 4. Nutritional Aspects and Prospects of the Food Use of *C. moschata*

### 4.1. Fruit Pulp

*Cucurbita moschata* is cultivated mainly for the production of fruits, which have high nutritional value. With versatile uses, the fruits of *C. moschata* are used in various dishes, such as salads, candy, cakes, pies, soups, and stews. In this context, [25] reported that the fruits of *C. moschata* have sensory/nutritional characteristics that make them attractive for consumption, especially the color of the fruit pulp, which varies from light yellow to intense orange, expressing wide variation (Figure 1). According to these authors, the carbohydrate content gives the pulp of *C. moschata* fruits a slightly sweet flavor, making it appealing for consumption. Additionally, studies show that *C. moschata* fruits have high levels of minerals such as K (42,194.000 mg/kg), Ca (6684.85 mg/kg), P (3040.48 mg/kg), Mg (1590.40 mg/kg), and Cu (8.44 mg/kg) [26].

Studies involving the analysis of *C. moschata* fruit pulp commonly report the observation of high levels of carotenoids [1,2,3,27], which allows *C. moschata* to be classified as a carotenogenic vegetable. These studies found that the carotenogenic profile in *C. moschata* fruits consists of approximately 19 carotenoids [1], with *β*- and *α*-carotene being predominant. In line with this, Carvalho et al. [1] described variation from 244.22 to 141.95 μg/g for *β*-carotene, and from 67.06 to 72.99 μg/g for *α*-carotene content; from the assessment of *C. moschata* landraces. As a result, *C. moschata* has been considered an important source of carotenoids, especially *β*-carotene, expressing higher levels of this component than found in other important carotenogenic vegetables, such as carrots [28].

*β*-carotene is known for its pronounced pro-vitamin A activity [29], in addition to its antioxidant activities [30]. In this sense, it is pertinent to mention that the pro-vitamin A activity of carotenoids is conditioned by the presence, in their structures, of ring forms called *β*-ionone, and by the presence of polyene chains [31]. With two *β*-ionone rings and one polyene chain, *β*-carotene has higher pro-vitamin A activity than other carotenoids with only one *β*-ionone ring, such as α-carotene and *β*-cryptoxanthin. Thus, the consumption of *C. moschata* fruit represents an important dietary source of vitamin A. In fact, *C. moschata* is one of the crops used in the Brazilian Biofortification Program (BioFORT) aimed at the biofortification of vitamin A precursors, led by EMBRAPA [32]. The combination of the fundamental characteristics of biofortification programs, such as high production potential and profitability, high efficiency in reducing micronutrient deficiencies in humans, and good acceptability by producers and consumers in cultivation regions [33], corroborate the choice of *C. moschata* as a strategic crop in biofortification programs aimed at overcoming vitamin A deficiency.

### 4.2. Seeds and Seed Oil

The production of *C. moschata* seed oil represents a promising alternative for the cultivation of this vegetable, owing to its high nutritional and physicochemical qualities. Composed of approximately 75% unsaturated fatty acids and with a high content of monounsaturated fatty acids such as oleic acid [5,6,7], the oil from *C. moschata* seeds is an excellent substitute for vegetable lipid sources with high levels of saturated fatty acids, which are harmful to human health. In addition, studies have reported that the seeds and seed oil of *C. moschata* contain high levels of antioxidant components such as vitamin E and carotenoids, which are beneficial to human health [8,9] and protect the oil against oxidative processes that can lead to rancidity.

In addition to the nutritional and physicochemical qualities of the oil from its seeds, *C. moschata* has high potential for seed and oil production. The lipid content represents up to 49% of the composition of *C. moschata* seeds [5,10], and studies with the germplasm of this vegetable have already identified accessions with seed productivity of up to 0.58 t ha^−1^ [11].

Despite their high nutritional value, most seeds produced from the cultivation of *C. moschata* are discarded [34], particularly in Brazil. Therefore, the production and consumption of *C. moschata* fruits are crucial for ensuring healthier diets and promoting food security in regions where they are cultivated, particularly in less developed regions and in the context of family farming. Additionally, as highlighted by [11], the use of *C. moschata* seeds for oil production represents a strategic alternative to complement the diet and increases the income of farmers involved in the production of this vegetable.

## 5. Breeding of Pumpkins and the Breeding Program of BGH-UFV

Worldwide, *C. pepo*, *C. maxima,* and *C. moschata* account for the majority of pumpkin production, serving a variety of purposes ranging from fruit production to the production of seed oil for human consumption. The production systems for these vegetables are often characterized by the use of unimproved local varieties [35], indicating the incipience of programs aimed at their breeding.

Pumpkin breeding programs often reflect the specific demands of each region. Corroborating this, in southwestern European countries, for example, pumpkin breeding programs, mostly for *C. pepo*, have prioritized the improvement of aspects related to seed oil production [36]. In these countries, pumpkin seed oil is produced mainly from the variety known as Styriaca or Styrian oil pumpkin, a variety resulting from a natural mutation that occurred in *C. pepo* in the mid-19th century in the province of Styria in southeastern Austria. Fruhwirth et al. (2008) described this variety as the result of the mutation of a single recessive gene that led to the formation of seeds with thin integuments or “naked seeds”, facilitating oil extraction [36].

The use of *C. pepo* var. styriaca has encouraged programs aimed at improving pumpkin seed oil in different regions of Europe, and as a result, numerous hybrids have been generated from crosses involving the Styriaca variety, aiming at greater seed yield and the introduction of resistance against pests and/or viruses that limit crop production [35,37]. These studies reported the production of hybrids with high oil content in the seeds (45–49%) and described the physicochemical aspects of the seed oil, reporting the predominance of linoleic acid.

Despite the high nutritional and physicochemical qualities of *C. moschata* seed oil, international breeding programs are still incipient compared to those for *C. pepo*. In this sense, studies on *C. moschata* have commonly focused on evaluating the native germplasm in different countries or regions to identify promising parents for improving the nutritional aspects of fruits and seed oil [11,35,38]. Therefore, breeding programs for *C. moschata* are commonly in the prebreeding phase. It is pertinent highlighting the recent development of hull-less *C. moschata* cultivars, as described by Kaur et al. [39]. These authors detail the successful attempt to transfer hull-less trait from *C. pepo* into *C. moschata* germplasm through conventional pollination and ovule culture, using four parents of hull-less *C. pepo* and six of hulled *C. moschata*. These authors highlight that the outcome of this study would pave the way for enhancing the productivity and multi-season cultivation of snack-seeded pumpkin even in subtropical and tropical regions.

In Brazil, the cultivation of *C. moschata* is primarily aimed at fruit production. Therefore, in addition to obtaining greater fruit productivity, breeding programs for this vegetable have prioritized the development of earlier cycle varieties and the production of fruits with higher organoleptic and chemical nutritional qualities, greater uniformity, and postharvest quality [40]. An example of this is the study by [41], involving the assessment of *C. moschata* progenies, which reported the identification of promising individuals for use in selecting for characteristics related to the chemical-nutritional quality of fruits, such as solute solids content, total carotenoid content, and *β*-carotene content.

More recently, in addition to achieving greater fruit productivity, pumpkin breeding programs have prioritized meeting demands, such as the development of earlier cycle varieties and the production of fruits with greater organoleptic and chemical-nutritional quality, greater uniformity, and greater postharvest quality [40,42,43,44]. In line with this, the UFV has been conducting a program aimed at genetic improvement for productivity and seed oil profiles of *C. moschata*. In the initial stages, this program assessed approximately 145 accessions of *C. moschata* maintained at the BGB-UFV [6,11]. Sobreira [6] identified the accessions BGH-7765, BGH-4615, and BGH-7319 as the most promising for seed oil production, with seed mass production per fruit ranging from 75 to 92 g and oleic acid content in seed oil ranging from 21 to 28%. In turn, [11] identified accessions with seed oil productivity of up to 0.27 t ha^−1^ and accessions with high oleic acid content in the oil, highlighting that these accessions are promising for use in breeding programs aimed at improving fatty acid profile and seed oil productivity. Figure 2 shows the cold extraction of seed oil from *C. moschata* and the high oil content of the seeds.

The breeding program of *C. moschata* at UFV used promising parents identified in the initial stage of the breeding program, as described by Laurindo et al. [45]. When using the parents BGH-7319 and BGH-7765 in crosses with “bush-type” pumpkin cultivars, namely Piramoita and Troco verde, the authors reported that the hybrid BGH-7319 × Piramoita stood out, with a seed mass production per fruit of 55g. In addition, they identified the ‘2 F_2_ population’ derived from the cross BGH-7319 × Tronco Verde as the most promising for obtaining plants with bush-type growth. Close to this, when evaluating generations resulting from the cross BGH-7765 × Tronco Verde, based on the factor analysis and ideotype-design (FAI-BLUP) selection index, under a selection intensity of 10%, Oliveira et al. [24] identified 25 genotypes with high potential for bush-type plants and high seed production, identifying accessions with seed mass per fruit of up to 82 g. Corroborating this, Table 1 shows the results obtained for the fatty acid profile of *C. moschata* germplasm from BGH-UFV and compares these results with other studies with *C. moschata*.

The UFV breeding program of *C. moschata* prioritized obtaining genotypes with shorter internodes and compact growth habit to reduce plant spacing and increase seed and oil productivity (Figure 3). This is linked to the fact that *C. moschata* has long internodes, associated with vigorous growth and indeterminate growth habit [25], which makes it difficult to adopt shorter spacing between plants and obtain higher seed and oil productivity. Thus, this breeding program has successfully emphasized the transfer of alleles related to compact plant size to *C. moschata* genotypes with a high potential for seed oil production [24,45,49].

It is worth mentioning that hybrid pumpkins from the Tetsukabuto segment, resulting from the cross between *C. maxima* × *C. moschata*, are the preferred choice in important consumer markets such as the Central-West, Southeast, and South regions of Brazil [40]. The preference for these hybrids stems from their superiority in terms of uniformity and postharvest fruit quality, which is associated with higher productivity compared to open-pollinated varieties. Therefore, in addition to obtaining greater fruit productivity, pumpkin breeding programs must prioritize the development of earlier cycle varieties that produce fruits with greater organoleptic and chemical nutritional qualities, greater uniformity, and postharvest quality.

It is also worth mentioning that Brazil imports large quantities of hybrid pumpkin seeds, which increases production prices. Based on a survey of the Brazilian National Cultivar Registry, 76 pumpkin hybrids resulting from the cross between *C. maxima* and *C. moschata* were identified, compared to 192 cultivars of *C. moschata* and 66 cultivars of *C. maxima*, demonstrating the reduced number of pumpkin hybrids developed in Brazil compared to the number of varieties [50].

## 6. Conclusions

The consumption of *C. moschata* fruits represents an important contribution of vitamin A to human nutrition, and the oil from its seeds is a lipid source with promising uses in human nutrition. The germplasm of *C. moschata* in Brazil shows remarkable variability, partly driven by its adaptation to the wide edaphoclimatic variations found in the country. As a result, the Brazilian germplasm of *C. moschata* is a strategic resource for the breeding of this vegetable and other species of the *Cucurbita* genus.

BGH-UFV has one of the largest collections of *C. moschata* in Brazil, resulting from a long period of collection throughout the Brazilian territory. Studies characterizing this germplasm have identified promising genotypes as sources of alleles for increasing the carotenoid content in the fruit pulp and oleic acid content in the seed oil.

The increase in carotenoid content, particularly that of components such as *β*-carotene, provides a greater supply of vitamin A for human nutrition. The increased oleic acid content in *C. moschata* seed oil confers high nutritional quality to the oil, making its use in human nutrition promising.

The breeding program of *C. moschata* conducted at the UFV has prioritized the production of compact growth habit genotypes to reduce plant spacing and increase seed and oil productivity. This study highlights innovative aspects in the production of *C. moschata*, such as the development of genotypes with compact growth habit and higher oil quality. The prospects for the *C. moschata* breeding program at BGH-UFV involve the continued use of this germplasm for the development of genotypes with compact growth habit and high seed oil quality.

Therefore, the production and consumption of *C. moschata* fruits and seed oil play important roles in promoting healthy diets and food security for populations that consume this vegetable.

## Figures and Tables

**Figure 1 plants-14-02317-f001:**
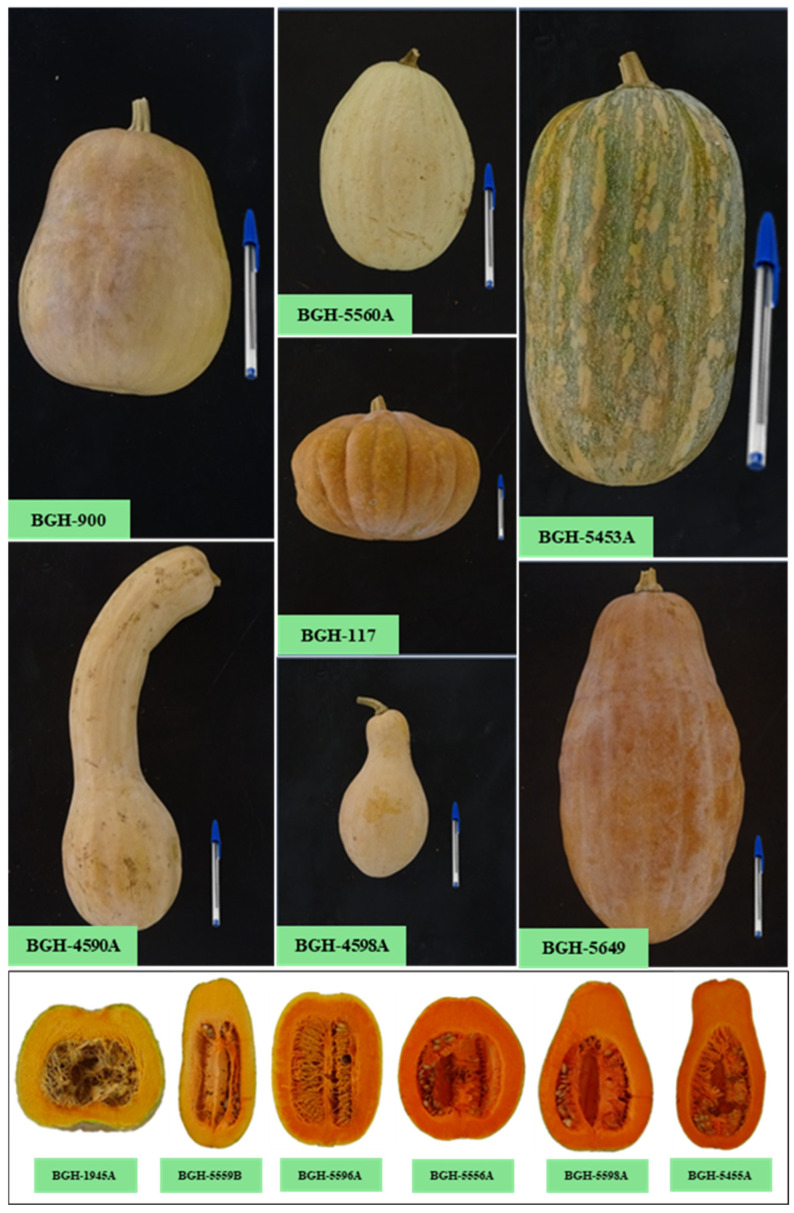
Morphological variability of fruits and variation in the color of fruit pulp between accessions of *C. moschata*. Source: personal archive of Gomes, R.S.

**Figure 2 plants-14-02317-f002:**
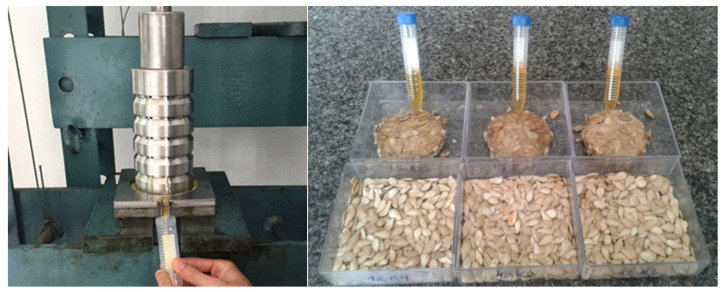
Cold extraction process of oil from *C. moschata* seeds. Source: personal archive of Gomes, R.S.

**Figure 3 plants-14-02317-f003:**
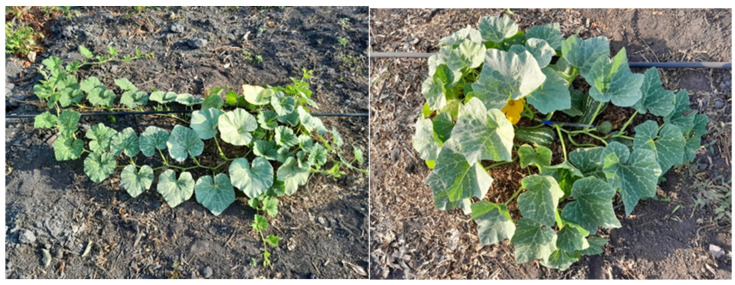
*Cucurbita moschata* genotype with long internodes (**left**), and genotype with short internodes and compact growth habit (**right**). Source: personal archive of Gomes, R.S.

**Table 1 plants-14-02317-t001:** Comparison between chemical-nutritional aspects of *C. moschata* seed oil of BGH-UFV with other *C. moschata* germplasm.

Studies With	Palmitic Acid 16:0 (%)	Stearic Acid 18:0 (%)	Oleic Acid 18:1(%)	Linoleic Acid 18:2 (%)	Linolenic Acid 18:3 (%)
BGH-UFV germplasm average [6]	15.07	9.09	21.15	54.46	0.21
BGH-UFV germplasm average [46]	15.16	9.60	24.55	50.66	0.18
BGH-UFV germplasm average [47]	14.35	9.96	27.70	48.20	-
Other studies with *C. moschata* germplasm [48]	16.64	6.66	23.07	51.58	0.54
Other studies with *C. moschata* germplasm [7]	15.56	9.71	29.50	43.6	0.14

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
