# Peer review of "Characterization, Conservation, and Breeding of Winter Squash (Cucurbita moschata Duchesne): Case Study of the Collection Maintained at the Federal University of Viçosa Vegetable Germplasm Bank"

_plants, 2025, doi:10.3390/plants14152317_

Round 1

Reviewer 1 Report

Comments and Suggestions for Authors

After a brief introduction, the domestication and genetic diversity of Cucurbita moschata is touched on even more briefly. The focus is then placed on the BGH-UFV collection. A wide range of topics is touched on (from ingredients to university breeding programmes), but nowhere is there any real in-depth coverage. Undoubtedly, an important resource is described that is worth publishing. However, it remains unclear what the exact purpose of this manuscript is.

The manuscript type review seems rather inappropriate. Although various studies conducted on C. moschata are listed, the focus is on activities of the BGH-UFV genebank. Due to the brevity of the manuscript, a genebank report (if this is possible) or a short communication would be appropriate.

Author Response

RESPONSE LETTER TO THE REVIEWER 1

Dear Editor, We are pleased to send you the responses regarding the corrections suggested by the reviewer 1.

We are glad upon the possibility of publishing this study on Plants.

We consider the suggestions provided by the reviewer very pertinent. We paid great attention to the suggestions and believe that this has greatly improved the current version of the manuscript.

Please see the responses to each aspect raised.

  • Reviewer 1 “After a brief introduction, the domestication and genetic diversity of Cucurbita moschata is touched on even more briefly. The focus is then placed on the BGH-UFV collection. A wide range of topics is touched on (from ingredients to university breeding programmes), but nowhere is there any real in-depth coverage. Undoubtedly, an important resource is described that is worth publishing. However, it remains unclear what the exact purpose of this manuscript is.

Answer: As can be seen in the structure of the manuscript, from its title to its topics, our main objective was to provide a review of the main studies involving the collection of C. moschata maintained at the Vegetable Germplasm Bank of the Federal University of Viçosa (BGH-UFV). These aspects are emphasized mainly in section 5 “Genetic breeding of pumpkins”. In this section, we address everything from the initial characterization of the germplasm to the use of part of this germplasm in breeding programs aimed at increasing seed oil productivity and reducing plant size, as exemplified by excerpts such as:

“In the initial stages, this program assessed approximately 145 accessions of C. moschata maintained at the BGB-UFV [6, 11]. [6] identified the accessions BGH-7765, BGH-4615, and BGH-7319 as the most promising for seed oil production, with seed mass production per fruit ranging from 75 to 92 g and oleic acid content in seed oil ranging from 21 to 28%.”

 At the same time, we understand that it is crucial that an article with this theme addresses aspects such as the domestication process of the species, conservation of its germplasm, and nutritional aspects of fruits and seeds.

Destacamos que tivemos atenção para que este artigo estivesse alinhado à temática da edição especial para qual o submetemos, a edição intitulada “Characterization and conservation of vegetable genetic resources”. Com isto, o artigo dá maior ênfase aos aspectos relacionados à caraterização e ao uso deste germoplasma em programas de melhoramento.

We would like to emphasize that we have taken care to ensure that this manuscript is aligned with the theme of the special issue to which we submitted it, the issue entitled “Characterization and conservation of vegetable genetic resources”. Therefore, the manuscript places greater emphasis on aspects related to the characterization and use of this germplasm in breeding programs.

Reviewer 1: “[...] but nowhere is there any real in-depth coverage.”

Answer: As already highlighted, we sought to emphasize aspects related to the characterization and use of germplasm. As a result, the article does not provide in-depth coverage of topics such as nutritional aspects. At the same time, we would like to emphasize that each topic provides comprehensive coverage of the most pertinent aspects. Corroborating this, we highlight the robustness of the references that support each of these topics. For example, the topic “Domestication and genetic diversity” is based on studies by researchers who are references on the subject, studies published in widely known journals for their high prestige, such as Science, as can be seen in the following excerpt:

“This evidence demonstrates that this species was present in Latin America before its colo-nization and appears to have been essential to the diet of the native people of this conti-nente, dating back to the regions of Colombia and Ecuador [14, 15, 16].”

A similar approach is found in the topic “nutritional aspects and prospects of the food use of C. moschata”. In this topic we were careful to address the most pertinent nutritional aspects - those that we have studied in the C. moschata collection at BGH-UFV, namely the nutritional aspects of the fruit pulp and seed oil. This topic is based on the most prestigious studies on the subject, which can be observed in the carotenoid content of the fruit pulp:

“Studies involving the analysis of C. moschata fruit pulp commonly report the observa-tion of high levels of carotenoids [1, 2, 3, 27], which allows C. moschata to be classified as a ca-rotenogenic vegetable.”

And about seed oil:

“Composed of approximately 75% unsaturated fatty acids and with a high content of mo-nounsaturated fatty acids such as oleic acid [5, 6, 7], the oil from C. moschata seeds is an excellent substitute for vegetable lipid sources with high levels of saturated fatty acids, which are harmful to human health.”

We emphasize that the contributions of the nutritional aspects of the fruit pulp and seed oil of C. moschata to human nutrition are based on highly prestigious studies in the area of ​​human nutrition, as can be seen in the following excerpts:

β-carotene is known for its pronounced pro-vitamin A activity [29], in addition to its antioxidant activities [30].”

“Composed of approximately 75% unsaturated fatty acids and with a high content of mo-nounsaturated fatty acids such as oleic acid [5, 6, 7], the oil from C. moschata seeds is an excellent substitute for vegetable lipid sources with high levels of saturated fatty acids, which are harmful to human health.”

Thus, although the manuscript does not provide in-depth coverage of each topic, it was carefully prepared to cover the most pertinent technical-scientific aspects of each topic, always based on studies that are references on the subject, published in journals with high academic prestige.

  • Reviewer 1 “The manuscript type review seems rather inappropriate. Although various studies conducted on moschata are listed, the focus is on activities of the BGH-UFV genebank. Due to the brevity of the manuscript, a genebank report (if this is possible) or a short communication would be appropriate.

Answer: We would very much like the manuscript to consist of a review. As previously explained, our main objective was to provide a review of the main studies involving the C. moschata collection maintained at BGH-UFV.

At the same time, we recognize the need to make the objective of the article clearer and more aligned with its structure. Therefore, we have made brief changes to the abstract to improve this aspect. Please check the changes identified by Track Changes in red in the abstract.

Reviewer 2 Report

Comments and Suggestions for Authors

Please use the correct Latin name for Cucurbita moschata, which is Cucurbita moschata Duchesne - it is because there were other taxonomists with D. as the initial, and therefore the full name of the taxonomist, which was Duchesne, is used for clarity. Be consistent throughout the whole manuscript.

Expression 'Genetic breeding' - for me, this expression is a bit vague, as breeding involves genetic principles, please consider 'breeding'

Please update section 2: check the recent publication on cucurbits domestication by Chomicki, G.; Schaefer, H.; Renner, S.S. Origin and domestication of Cucurbitaceae crops: insights from phylogenies, genomics and archaeology. New Phytol. 2019, 226, 1240–1255.

Figure 1. Is it possible to provide the names of the Horticultural groups to which the fruits of the presented accessions belong? Is it possible to refer to Amy Goldman's Book 2004 or Brazilian types of C. moschata cultivars? 

Section 3: What about the genetic diversity of the collection? Were there any studies at the DNA level performed, or are there any studies considered? 

Section 4: The levels of minerals are provided - please also provide the levels of major carotenoids in the pulp, for consistency 

Section 5: Regarding breeding objectives - any comments on virus resistance? 

References:  please carefully check and format the references according to the journal's requirements, also check out order names and initials because there are errors (for example, reference 35 - not Ercolanob but Ercolano, etc.) 

Please check carefully the manuscript for editorial errors or inconsistencies. Here are some examples: 

Line 18: Ccucurbita - Cucurbita

Line 30: C. moschata - italic

Line 82-83: C. moschata - italic

Line 87: addi-tion - addition

Line 94-95: C. moschata - italic

Line 132: - In line with this ... - In line with this, Gomes et al 2020 [11]

Line 134: impor-tant - important

Line 140: In this context, [25] ... - In this context, Gomes et al 2020 [25]

Lines 146-147: please the levels of K (42,194,000 mg/kg), Ca (6,684,85 mg/kg), P (3,040,48 mg/kg), Mg (1,590,40 mg/kg), and Cu (8,44 mg/kg) - looks unusually high

Line 150: ca-rotenogenic - carotenogenic

Line 163: vita-min - vitamin

Line 170: C. moschata - italic

Line 207: C. pepo var. Styriaca = C. pepo var. styriaca

Line 216: tudies - studies

lines 214-219: Please consider to discuss what are perspectives of hull-less C.moschata cultivars development, as recently naked seed/hull-less seed genotypes were described it in C. moschata

Line 250: with a seed mass production per fruit of 55.18 g - please consider 55 g

Line 251: ‘2 F2 population’ - it is not clear 

Lines 251 and 270: use multiplication sign ×, not x, in the formulas of the crosses

Line 268: compact size - consider compact growth habit

Author Response

RESPONSE LETTER TO THE REVIEWER 2

Dear Editor, We are pleased to send you the responses regarding the corrections suggested by the reviewer 2.

We are glad upon the possibility of publishing this study on Plants.

We consider the suggestions provided by the reviewer very pertinent. We paid great attention to the suggestions and believe that this has greatly improved the current version of the manuscript.

Please see the responses to each aspect raised.

  • Reviewer 2 “Please use the correct Latin name for Cucurbita moschata, which is Cucurbita moschata Duchesne - it is because there were other taxonomists with D. as the initial, and therefore the full name of the taxonomist, which was Duchesne, is used for clarity. Be consistent throughout the whole manuscript.”

Answer: We found the suggestion to be pertinent. We have added the name of the taxonomist in all instances where the scientific name is cited in full. Please note the changes identified by Track Changes in red. We would like to point out that in most cases we refer to the species by the abbreviated form of its name, namely C. moschata.

  • Reviewer 2 “Expression 'Genetic breeding' - for me, this expression is a bit vague, as breeding involves genetic principles, please consider 'breeding'.

Answer: We agree with the suggestion. Please note the changes identified by Track Changes in red.

  • Reviewer 2 “Please update section 2: check the recent publication on cucurbits domestication by Chomicki, G.; Schaefer, H.; Renner, S.S. Origin and domestication of Cucurbitaceae crops: insights from phylogenies, genomics and archaeology. New Phytol. 2019, 226, 1240–1255.”

Answer: We carefully read the aforementioned study (Chomicki et al. 2019), which provides extensive research on the phylogeny, genomics, and archaeology of species in the Cucurbitacea family. Regarding C. moschata, the study by Chomicki et al. 2019 cites other studies that corroborate the information we provide in our review article. Thus, although the study by Chomicki et al. 2019 is a major piece of work, it does not provide additional information on the domestication and genetic diversity of C. moschata.

  • Reviewer 2 “Figure 1. Is it possible to provide the names of the Horticultural groups to which the fruits of the presented accessions belong? Is it possible to refer to Amy Goldman's Book 2004 or Brazilian types of C. moschata cultivars?”

Answer: We have inserted a brief text in the figure caption, detailing the cultivar group to which the fruits belong. Please note the changes identified by Track Changes in red.

  • Reviewer 2 “Section 3: What about the genetic diversity of the collection? Were there any studies at the DNA level performed, or are there any studies considered? 

Resposta: inserimos um breve texto detalhando estas informações. Houve uma pequena alteração na estrutura deste parágrafo e do parágrafo seguinte. Por favor observar as alterações identified by Track Changes in red entre as linhas 134 e 140.

Answer: We have inserted a brief text detailing this information. There has been a slight change in the structure of this paragraph and the following paragraph. Please note the changes identified by Track Changes in red between lines 147 and 152.

  • Reviewer 2 “Section 4: The levels of minerals are provided - please also provide the levels of major carotenoids in the pulp, for consistency”

Answer: We have added a brief paragraph detailing this information. Please note the changes identified by Track Changes in red between lines 174 and 176.

  • Reviewer 2 “Section 5: Regarding breeding objectives - any comments on virus resistance?”

Answer: Currently, the pumpkin breeding program linked to BGH-UFV does not include breeding for virus resistance. However, initially, part of the BGH-UFV C. moschata collection was assessed for virus resistance. We cite throughout the manuscript the study by Moura et al. (2005), who undertook the assessment of a large sample of the BGH-UFV C. moschata collection for resistance to viruses such as Zucchini yellow mosaic virus (ZYMV). Please check the study citation in line 131.

Answer: Currently, the pumpkin breeding program linked to BGH-UFV does not include breeding for virus resistance. However, initially, part of the BGH-UFV C. moschata collection was assessed for virus resistance. We cite throughout the manuscript the study by Moura et al. (2005), who undertook the assessment of a large sample of the BGH-UFV C. moschata collection for resistance to viruses such as Zucchini yellow mosaic virus (ZYMV). Please check the study citation in line 144.

  • Reviewer 2 “References: please carefully check and format the references according to the journal's requirements, also check out order names and initials because there are errors (for example, reference 35 - not Ercolanob but Ercolano, etc.)”.

Answer: We have reviewed it carefully and made the necessary corrections. Please note the changes identified by Track Changes in red.

  • Reviewer 2 “Please check carefully the manuscript for editorial errors or inconsistencies. Here are some examples: Line 18: Ccucurbita – Cucurbita”.

Answer: the correction was made. Please see line 19.

  • Reviewer 2 “ Line 30: C. moschata – italic”.

Answer: the correction was made.

  • Reviewer 2 “Line 82-83: C. moschata – italic”.

Answer: the correction was made.

  • Reviewer 2 “Line 87: addi-tion – addition.”

Answer: the correction was made.

  • Reviewer 2 “Line 94-95: C. moschata – italic.”

Answer: the correction was made.  Please see the lines 113 and 115.

  • Reviewer 2 “Line 132: - In line with this ... - In line with this, Gomes et al 2020 [11].”

Answer: the correction was made. Please see lines 153 and 154.

  • Reviewer 2 “Line 134: impor-tant – importante.”

Answer: The correction was made.

  • Reviewer 2 “Line 140: In this context, [25] ... - In this context, Gomes et al 2020 [25].”

Resposta: The correction was made. Please see the line 149.

Answer: The correction was made. Please see line 162.

  • Reviewer 2 “Lines 146-147: please the levels of K (42,194,000 mg/kg), Ca (6,684,85 mg/kg), P (3,040,48 mg/kg), Mg (1,590,40 mg/kg), and Cu (8,44 mg/kg) - looks unusually high.”

Answer: We have carefully analyzed the study cited. The values ​​correspond exactly with those cited in the study. We would like to point out, however, that the levels are per kilogram, which may give the impression that they are very high.

  • Reviewer 2 “Line 150: ca-rotenogenic – carotenogenic.”

Answer: The correction was made.

  • Reviewer 2 “Line 163: vita-min – vitamin.”

Answer: The correction was made.

  • Reviewer 2 “Line 170: C. moschata – italic.”

Answer: The correction was made.

  • Reviewer 2 “Line 207: C. pepo var. Styriaca = C. pepo var. styriaca.”

Answer: The correction was made. Please see line 231.

  • Reviewer 2 “Line 216: tudies – studies.”

Answer: The correction was made. Please see the line 240.

  • Reviewer 2 “lines 214-219: Please consider to discuss what are perspectives of hull-less C.moschata cultivars development, as recently naked seed/hull-less seed genotypes were described it in C. moschata.”

Answer: We found it pertinent to add the information. With this, we have added the study by Kaur et al (2023). Please see the lines 243 to 249. With the addition of this reference, there were changes in the numbering of subsequent references. Please see the changes identified by Track Changes in red.

  • Reviewer 2 “Line 250: with a seed mass production per fruit of 55.18 g - please consider 55 g.”

Answer: The correction was made. Please see the line 280.

  • Reviewer 2 “Line 251: ‘2 F2 population’ - it is not clear.”

Answer: As the authors detail in their study, they obtained 3 F2 populations: ‘Population 1 F2’, ‘Population 2 F2’ and ‘Population 3 F2’, from the hybrids BGH-7319 × Piramoita, BGH-7319 × Tronco Verde and BGH-7765 × Tronco Verde, respectively (Laurindo et al. 2020). Thus, the denotation 2 F2 refers to a specific self-fertilization population, in the case, a self-fertilization second poulation from 3 population that were assessed.

  • Reviewer 2 “Lines 251 and 270: use multiplication sign ×, not x, in the formulas of the crosses.”

Answer: The correction was made. Please see the line 280 to 283.

  • Reviewer 2 “Line 268: compact size - consider compact growth habit.”

Answer: The correction was made. Please see line 333.

Reviewer 3 Report

Comments and Suggestions for Authors

A review of breeding of C. moschata at the Vegetable Germplasm Bank of the Federal University of Viçosa (BGH-UFV) in Brazil. The primary focus is on oil quality. Comparisons are made to  varieties developed from other species of Cucurbita. The paper would be GREATLY enhanced with the addition of a Table that makes comparisons of oil quality and other important characteristics relating to pumpkin breeding enabling comparisons with varieties developed in these other species and with selections made already using C. moschata. A key paper for the authors to examine is: J. Agric. Food Chem. 2007, 55, 4005−4013 4005
 Oil and Tocopherol Content and Composition of Pumpkin Seed
 Oil in 12 Cultivars
 DAVID G. STEVENSON,*,² FRED J. ELLER,³ LIPING WANG,§ JAY-LIN JANE,§
 TONG WANG,§ AND GEORGE E. INGLETT²

Author Response

RESPONSE LETTER TO THE REVIEWER 3

Dear Editor, We are pleased to send you the responses regarding the corrections suggested by the reviewer 3.

We are glad upon the possibility of publishing this study on Plants.

We consider the suggestions provided by the reviewer very pertinent. We paid great attention to the suggestions and believe that this has greatly improved the current version of the manuscript.

Please see the responses to each aspect raised.

  • Reviewer 3 “A review of breeding of C. moschata at the Vegetable Germplasm Bank of the Federal University of Viçosa (BGH-UFV) in Brazil. The primary focus is on oil quality. Comparisons are made to varieties developed from other species of Cucurbita. The paper would be GREATLY enhanced with the addition of a Table that makes comparisons of oil quality and other important characteristics relating to pumpkin breeding enabling comparisons with varieties developed in these other species and with selections made already using C. moschata. A key paper for the authors to examine is: J. Agric. Food Chem. 2007, 55, 4005−4013 4005 Oil and Tocopherol Content and Composition of Pumpkin Seed Oil in 12 Cultivars

DAVID G. STEVENSON,*,² FRED J. ELLER,³ LIPING WANG,§ JAY-LIN JANE,§ TONG WANG,§ AND GEORGE E. INGLETT².”

Answer: We found the suggestion pertinent and therefore added a table with this information. Please see the table between lines 300 and 319. We highlight that there were some changes in the references due to the addition of 2 new references in the table. Please see the changes identified by Track Changes in red.

Reviewer 4 Report

Comments and Suggestions for Authors

Dear Authors,

Your manuscript entitled „Characterization, conservation, and genetic breeding of winter squash (Cucurbita moschata D.): case study of the collection maintained at the Federal University of Viçosa Vegetable Germplasm Bank” contains interesting results. Nevertheless, I have found some imperfections, which (in my opinion) sould be improved or at least clarified before an eventual publication. I have listed them below;

  1. Abstract section should reffer to all sections of manuscript. Please add the main data concerning methods and main conclusions.
  2. In my opinion the characteristics of Winter squash (Cucurbita moschata D). should be enlarged. Information about lifespan, life form, origin and native range, nutritional value, use should be given in Introduction. Moreover, the reason of undertaking presented investigations should be stronger justified.
  3. The chapter Material and methods is strongly desired. How the literature sources were find? Did You brownse the dtatabases (e.g., WoS, Scopus, Google Scholar Internet engine)? If so, which keywords were applied? I encourage You to visit page http://www.prisma-statement.org/ presenting PRISMA statements and charts depicting way of article search.

The detailed description of procedure of publications search will avoid impression, that some papers could have been omitted.

  1. In my opinion the Discussion chapter might be added. The presented results should be compared with review articles reffering to other useful alimentary plant species.
  2. The conclusions chapter should point out the novelty of Your findings and indicate the direcions of future studies.
  3. The way of miting literature sources in References chapter should be corrected according to Guide for Authors.

Author Response

RESPONSE LETTER TO THE REVIEWER 4

Dear Editor, We are pleased to send you the responses regarding the corrections suggested by the reviewer 4.

We are glad upon the possibility of publishing this study on Plants.

We consider the suggestions provided by the reviewer very pertinent. We paid great attention to the suggestions and believe that this has greatly improved the current version of the manuscript.

Please see the responses to each aspect raised.

  • Reviewer 4 “Abstract section should reffer to all sections of manuscript. Please add the main data concerning methods and main conclusions.”

Answer: We find the suggestion pertinent and have therefore added the main conclusions to the abstract. Please see the changes identified by Track Changes in red.

We highlight that the journal's rules determine that the abstract must have a maximum of 250 words. We emphasize the objective of the study, mentioning “This study aims reviewing the main aspects related to the characterization, conservation, and genetic breeding of C. moschata, emphasizing the studies with C. moschata accessions maintained by the Vegetable Germplasm Bank of the Federal University of Viçosa (BGH-UFV).” With this, we convey the idea that the study is based on a review.

  • Reviewer 4 “In my opinion the characteristics of Winter squash (Cucurbita moschata D). should be enlarged. Information about lifespan, life form, origin and native range, nutritional value, use should be given in Introduction. Moreover, the reason of undertaking presented investigations should be stronger justified.

Answer: We believe it is appropriate to add information by providing a brief description of the crop's botanical aspects. Please see lines 51 to 54.

We would like to highlight that the introduction covers the crop's nutritional aspects. Please check sections such as:

“The fruit pulp is rich in carotenoids, such as α and β-carotene [1, 2, 3], which are the main precursors of vitamin A and have pro-nounced antioxidant activity. Additionally, it is an excellent source of minerals, such as K, Ca, P, Mg, and Cu [3, 4].”

“The oil from its seeds contains appro-ximately 75% unsaturated fatty acids with a high content of oleic acid, a monounsatura-ted fatty acid [5, 6, 7]. Therefore, the seed oil of C. moschata constitutes an excellent substitute for lipid sources that are harmful to human health, such as those with high levels of saturated fatty acids.”

We highlight that more detailed information on the nutritional aspects of the crop is provided in topic 4 “4. Nutritional aspects and prospects of the food use of C. moschata.”

  • Reviewer 4 “The chapter Material and methods is strongly desired. How the literature sources were find? Did You brownse the dtatabases (e.g., WoS, Scopus, Google Scholar Internet engine)? If so, which keywords were applied? I encourage You to visit page http://www.prisma-statement.org/ presenting PRISMA statements and charts depicting way of article search.

Answer: As can be seen in the structure of the manuscript, from its title to its topics, our main objective was to provide a review of the main studies involving the collection of C. moschata maintained at the Vegetable Germplasm Bank of the Federal University of Viçosa (BGH-UFV). Thus, we have provided a review of the main studies with the germplasm of C. moschata from BGH-UFV, such as:

Sobreira F. M. Divergência genética entre acessos de abóbora para estabelecimento de coleção nuclear e pré-melhoramento para óleo funcional. D. Sc. Thesis, Universidade Federal de Viçosa. 2013. Available from: https://www.locus.ufv.br/handle/123456789/1367.

Gomes, R.S.; Machado Júnior, R.; Almeida, C.F.; Chagas, R.R.; Oliveira, R.L.; Delazari, F.T.; Silva, D.J.H. Brazilian germplasm of winter squash (Cucurbita moschata D.) displays vast genetic variability, allowing identification of promising genotypes for agro-morphological traits. Plos One. v. 15, p. 1-16, 2020. Doi: 10.1371/journal.pone.0230546

Gomes, R.S.; Machado Júnior, R.; Almeida, C.F.; Oliveira, R.L.; Chagas, R.R.; Pereira, E.D.; Delazari, F.T.; Da Silva, D.J.H. Identification of high seed oil yield and high oleic acid content in Brazilian germplasm of winter squash (Cucurbita moschata D.). Saudi J. Biol. Sci. v. 29, p. 2280- 2290. 2022. Doi: 10.1016/j.sjbs.2021.11.064

Almeida, C.F.; Gomes, R.S.; Machado Junior, R.; Oliveira, R.L.; Nardino, M.; Da Silva, D.J.H. Inheritance of traits related to yield and fatty acid profile of winter squash seed oil. Sci. Hortic-Amsterdam. v. 308, p. 1-10. 2023. Doi: 10.1016/j.scienta.2022.111523

  • Reviewer 4 “In my opinion the Discussion chapter might be added. The presented results should be compared with review articles reffering to other useful alimentary plant species.”

Answer: We sought to emphasize aspects related to the characterization and use of germplasm. As a result, the manuscript does not provide in-depth coverage of topics such as nutritional aspects. At the same time, we would like to emphasize that each topic provides comprehensive coverage of the most pertinent aspects. In support of this, we would like to highlight the robustness of the references that support each of these topics. For example, the topic “4. Nutritional aspects and prospects of the food use of C. moschata” is based on studies by researchers who are references on the subject, studies published in widely known journals due to their high prestige, as can be seen in the following excerpt:

“Studies involving the analysis of C. moschata fruit pulp commonly report the observa-tion of high levels of carotenoids [1, 2, 3, 27], which allows C. moschata to be classified as a ca-rotenogenic vegetable.”

And about seed oil:

“Composed of approximately 75% unsaturated fatty acids and with a high content of mo-nounsaturated fatty acids such as oleic acid [5, 6, 7], the oil from C. moschata seeds is an excellent substitute for vegetable lipid sources with high levels of saturated fatty acids, which are harmful to human health.”

Thus, although the manuscript does not provide in-depth coverage of each topic, it was carefully prepared to cover the most pertinent technical-scientific aspects of each topic, always based on studies that are references on the subject, published in journals with high academic prestige.

  • Reviewer 4 “The conclusions chapter should point out the novelty of Your findings and indicate the direcions of future studies.”

Answer: We think it is pertinent to add the suggested aspects. Please see lines 365 to 368.

  • Reviewer 4 “The way of miting literature sources in References chapter should be corrected according to Guide for Authors.”

Answer: We carried out a careful analysis in preparing the reference, meeting the journal's standards.

Round 2

Reviewer 4 Report

Comments and Suggestions for Authors

Dear Authors,

Your manuscript has been sufficiently corrected. therefore I do not have any further remarks.

Author Response

RESPONSE LETTER TO ACADEMIC EDITOR COMMENTS

Dear Editor, We are pleased to send you the responses regarding your comments.  

We are glad upon the possibility of publishing this study on Plants.

We paid great attention to the suggestions and believe that this has greatly improved the current version of the manuscript.

Please see the responses to each aspect raised.

  • Academic editor. “Comment regarding this reply: The mentioned references are relatively old: 14 (1976), 15 (2003), 16 (2007). I suggest including some more recent pertinent references to this topic, such as: Dhatt, A. S., Pandey, S., Garcha, K. S., Verma, N., Sagar, V., & Sharma, M. (2024). Comprehensive review of pumpkin (Cucurbita spp.): Domestication, global distribution, genetic characterization, breeding strategies, and genomic insights. Vegetable Science, 51(02), 196-210. Lira, R., Eguiarte, L., Montes, S., Zizumbo-Villarreal, D., Marín, P. C. G., & Quesada, M. (2016). Homo sapiens–Cucurbita interaction in Mesoamerica: domestication, dissemination, and diversification. In Ethnobotany of Mexico: interactions of people and plants in Mesoamerica (pp. 389-401). New York, NY: Springer New York. Nyu, A. (2024). The Domestication of Pumpkins: Historical Perspectives and Modern Genetic Evidence. International Journal of Horticulture”.

Answer: Dear Editor, we found your suggestion pertinent. We have deleted the references to Whitaker et al. 1976 and Dillehay et al. 2007. We have added most of the references you suggested in this context, namely those of Lira et al. 2016 and Dhatt et al. 2024.

Although old, the study by Piperno et al. 2003 is a fundamental study in this context. Corroborating this, the study by Lira et al. 2016 is based on the study by Piperno et al. 2003. Therefore, we believe it is essential to maintain the reference to Piperno et al. 2003.

The study by Nyu, A. (2024), The Domestication of Pumpkins: Historical Perspectives and Modern Genetic Evidence. International Journal of Horticulture, does not provide details on the domestication process of C. moschata, therefore we did not add this reference.

Please see the changes:

Reference to Whitaker et al. 1976 has been deleted, see line 422.

Reference to Dillehay et al. 2007 has been deleted, see line 426.

Reference to Lira et al. 2016 has been added, see line 428.

Reference to Dhatt et al. 2024 has been added, see line 431.

Please note that the reference to Piperno et al. 2003 has been changed to number 14.

  • Academic editor. “Figure 1. Morphological variability of fruits and in the color of fruit pulp between accessions of C. moschata. Source: personal archive of Gomes, R.S. All accessions shown belong to the group of drie pumpkin cultivars; a group known in the Brazilian market for its large fruits, weighing up to 15 kg, fruits with thick skin and firm pulp. The explanatory sentence added in red color to Figure 1 should be part of the main text, not the figure legend. What is meant by “drie pumpkin”?

Answer: We made the change and with this the explanatory sentence is now part of the main text just below figure 1. Please see the change in line 111.

The term "drie pumpkin" refers to a specific group of pumpkins, according to the classification used in the Brazilian market. This is a well-known classification in the country. We understand that this term may be restricted to Brazil, so we have added a brief explanation in relation to this term. Please see the text on line 111.

  • Academic editor “L231-232: [36] described this variety as the result of the mutation of a single recessive gene that led to the formation of seeds with thin integuments or “naked seeds,” facilitating oil extraction. It has been noted that the authors' names of Gomes et al. have been overemphasized throughout the text (during the review process added in red color), while this sentence highlighted in lines 231-232, starts with a reference number only, where it would be opportune to start the sentence with the author’s name, adding the reference number in brackets”.

          Answer: Yes, in this way, the name "Gomes et al." was overemphasized. In part, the emphasis on "Gomes et al." resulted from a suggestion from reviewer 2, based on the first review of the article. In the first review, we followed the reviewer's suggestion, as can be seen in the Authors' Responses to Reviewer's Comments (Reviewer 2). Now, we find it appropriate to put some of the references to "Gomes et al." in number format. Please see all the changes in this context in red.

We have included the authors' names in the cases we found pertinent. Please see lines 175, 229, and 245.

  • Academic editor. “L289: Oiliveira et al. [24] identified 25 genotypes The author's name „Oiliveira” appears to be misspelt. Typos Carefully review and correct typos throughout the text (see, for example, the following, but there are others throughout the text): L84: The re-sults – change to results L177-179: In line with this, Carvalho et al. [1] describe variation from 244.22 to 141.95 μg/g for β-carotene, and from 67.06 to 72.99 μg/g for α-carotene contente; from the assessment of C. moschata ladraces”.

Answer: We identified the error and made the correction. Please see line 285. We carefully analyzed the text and corrected the misspelled mistakes
